# A New Filtering System for Using a Consumer Depth Camera at Close Range

**DOI:** 10.3390/s19163460

**Published:** 2019-08-08

**Authors:** Yuanxing Dai, Yanming Fu, Baichun Li, Xuewei Zhang, Tianbiao Yu, Wanshan Wang

**Affiliations:** 1School of Mechanical Engineering and Automation, Northeastern University, Shenyang 110004, China; 2Laboratory Management Center, Shenyang Sport University, Shenyang 110102, China; 3College of Aeronautical Engineering, Civil Aviation University of China, Tianjin 300000, China

**Keywords:** depth image filtering, point clouds filtering, Kinect v2, depth resolution, close range, hand pose

## Abstract

Using consumer depth cameras at close range yields a higher surface resolution of the object, but this makes more serious noises. This form of noise tends to be located at or on the edge of the realistic surface over a large area, which is an obstacle for real-time applications that do not rely on point cloud post-processing. In order to fill this gap, by analyzing the noise region based on position and shape, we proposed a composite filtering system for using consumer depth cameras at close range. The system consists of three main modules that are used to eliminate different types of noise areas. Taking the human hand depth image as an example, the proposed filtering system can eliminate most of the noise areas. All algorithms in the system are not based on window smoothing and are accelerated by the GPU. By using Kinect v2 and SR300, a large number of contrast experiments show that the system can get good results and has extremely high real-time performance, which can be used as a pre-step for real-time human-computer interaction, real-time 3D reconstruction, and further filtering.

## 1. Introduction

The reasons for the success of consumer depth cameras are low price, acceptable accuracy, lower learning costs, extensive applicability, and excellent portability. It has been applied in the fields such as body and facial recognition, 3D motion capture, and has been developing very rapidly.

Most of the depth cameras are based on time-of-flight principle, such as Kinect v2 and SR300. It can collect the laser spots array reflected by surfaces, and works out the time difference between emission and reflection to get the distance array of the scene [1,2]. Generally, the array is expressed as a gray image, and the gray value of the pixel is generated by the depth value of the position according to certain rules.

According to the principle of perspective, within the unit area, the closer the surface is to the camera, the more laser points will be reflected, which means that higher measurement point density will result in higher surface resolution. Although the range could be changed by using Draelos’s method [3], according to our observation, when a target surface gets close to the nearest limit, for example, 3D reconstruction of small objects [1], in the point cloud acquired by consumer depth cameras, there will be some irregular shape of noise areas surrounding or on the edge of the realistic surface (a surface consisting of laser points reflected by a real object, it is used to distinguish unrealistic surfaces formed by noise points that should not exist).

In the process of generating point clouds with depth cameras, we found that the closer the distance is, the more significant the phenomenon is. Figure 1 shows this phenomenon by using the depth images of a human hand at different distances. In static applications, these low confidence noise areas could be filtered effectively in a post-processing stage [4]. However, any time consumption is undesirable for real-time interactive applications [5,6,7].

Using a consumer depth camera at close range is a double-edged sword. Developers often aim to use depth images or video stream for further development, such as reverse engineering [8], human pose recognition [9,10,11,12], and 3D scene reconstruction [13]. In order to obtain a pure depth image with high accuracy and low noise, one option is to select expensive, high learning cost, and precise optical equipment (3D time-of-flight (ToF) camera or LIDAR). However, for time and money savings, an easier way is to place the object closer to get a higher resolution on the surface of the object. In this case, how to eliminate the noise deterioration caused by close-range use has become a problem that must be solved.

Traditional methods for reducing or eliminating color image noise are usually based on window smoothing or sharpening, such as median filtering [14], non-local means filter [15], bilateral filtering [14,16], joint bilateral filtering [17], etc. The principle is to make a window for each pixel in the image, and update the center pixel according to the value of other pixels in the window.

Different filters use different window selection methods and updated strategies [18]. However, the unmodified algorithm transplantation is not very suitable for depth image filtering. For edge noise of a depth image, joint bilateral filtering with reliable sources (usually from color images) can perform very well. It can better preserve the edge details of an object [19], but for human hand depth image filtering it also involves the lighting conditions [20,21,22] and the color difference of the foreground and background [23,24].

To fill this gap, in this article we proposed a composite filtering system for eliminating low confidence noise areas around or on the realistic surface and obtaining relatively pure point clouds of a human hand within close distance. In order to maximize the retention of raw data for further use, the system does not use a smoothing filtering algorithm. All the algorithms in the system are implemented by GPU-assisted parallel computing, thus making the system achieve very high real-time performance. Finally, the experiment results show that the proposed filtering system could eliminate the vast majority of noise areas. 

## 2. Noise Characterization

Severely distorted noise of a close-range collected depth image tends to concentrate on the area of the image where the depth gradient is large. For a highly integrated device, the calibration method for laser scanner cannot be used [25], and as a result, the user cannot adjust the data generation process in most cases (it depends on the camera, SR300 could be allowed to adjust laser intensity and type of built-in filters), but only using the deep data acquired from the device. Hence an in-depth understanding of the noise characteristics within the depth images is the first task to build an effective filtering system.

The noise in the depth image is actually the sum of spatial noise and temporal noise. The former can be construed as an inaccurate depth measurement, which mainly includes the zero depth (ZD) that cannot be measured (like NaN [26]), and the wrong depth (WD) that is far from the actual depth value. The latter refers to the phenomenon that the measured value fluctuates with time when the depth of this point does not change, thus, multi-frames are needed to eliminate the temporal noise [27] that may cause input delay to the interface system. More detailed elaborations are made in [26,27,28,29].

For real-time interactive applications, any delay should be avoided, then the best way is to start with spatial noise and eliminate the low confidence WD areas. Therefore, in this section, according to shape and location, the noise areas that seriously affect the correctness of point cloud generation are of the hand surface classified into three types. Two of them are original noise which are shown in Figure 2, and another one is residual noise which will be described in Section 3.3. 

Outlier noise. This is the point with WD that exists away from the realistic surface, and is usually randomly distributed in the depth image spatially and temporally, which usually has impacts on the pass-through filtering [30].Edge noise. This kind of noise exists regionally. It can be an unrealistic surface composed of noise points surrounds the realistic surface, or it can be part of the edge of a realistic surface with WD. The closer to the blank area, the greater the depth gradient. It would eventually point to the $z_{+}$ or $z_{-}$ (the positive or negative direction of the depth value).Plaque noise. This is a kind of residual noise. The filtering system may miss some plaque areas after filtering the first two kinds of noise. Most of the residual plaque areas are isolated and a few are connected to realistic surfaces.

Together, these three types of noise constitute a low confidence noise area in depth images. It is worth noting that the characteristics of the first two noises are not independent of each other, if the gradient in the window is too large, the extreme points in the same window could be regarded as outliers. Therefore, in the next section, a composite filtering system will be proposed for these types of noise.

## 3. Proposed Filtering System

Based on the analysis of the noise types in the depth image acquired in close range, different noise characteristics make it difficult for a single filter to perform well. Therefore, a filtering system consisting of multiple detection modules is proposed in Figure 3. The CPU only needs to obtain the depth image from the device, and obtain the filtered point cloud data from the GPU and display them. All filtering algorithms are running in GPU, the calculation part such as standard deviation (SD) calculation, depth to 3D coordinate conversion follows the calculate-when-using principle to reduce the read and write frequency of the graphics card memory. By setting a reasonable number of loops $n_{loop}$, it could eliminate most of the low confidence areas, and preserve realistic surfaces. Highly parallelized algorithms in the filtering system could save the computing resources of CPU and make the system run in real time. 

### 3.1. Improved Dixon Test

The outliers seriously interfere with the depth value-based hand region truncation, often causing truncation failure. As for the adjacent points belonging to a same realistic surface, their depth values are usually very close. For any non-zero point (NZP) $p\in I_{D}$, both value range and the SD of δ(p) could not be too large, where $\delta \left(p\right)\subset I_{D}$ is the neighbor set of p.

The central idea of the Dixon test is to determine whether extreme points are outliers by calculating the ratio between extreme point deviation and sample range. Equation (1) shows the Dixon outlier test method for up to 10 samples [31], where the $Q_{u}$ and $Q_{l}$ are used to identify the maximum and minimum sample respectively. $x_{1}$ and $x_{n}$ are the extremes of the arranged samples, and the values of $Q_{u}$ and $Q_{l}$ could reflect how large the gap is. Different confidence levels (α) correspond to different limits of $Q_{u}$ and $Q_{l}$, it can be obtained by looking up the table [31].
(1)$$\{\begin{array}{l} 3<n<7:Q_{u}=\frac{x_{n}-x_{n-1}}{x_{n}-x_{1}},\;Q_{l}=\frac{x_{2}-x_{1}}{x_{n}-x_{1}}\\ 8\leq n\leq 10:Q_{u}=\frac{x_{n}-x_{n-2}}{x_{n}-x_{2}},\;Q_{l}=\frac{x_{2}-x_{1}}{x_{n-1}-x_{1}} \end{array},\;x_{1},\ldots ,x_{n}\in \delta \left(p\right)$$

However, to some extent, the ratio only reflects the deviation between the extreme point and cluster of other points spatially. Therefore, there are natural defects in applying it directly to outlier detection in depth images. The reason is that it cannot reflect the discreteness of depth values of all points in the window macroscopically. However, the SD, which can reflect the dispersion on the value of the samples, cannot microscopically reflect the positional relationship between each point and the cluster. Figure 4 and Algorithm 1 show the improvements of the Dixon test. There are three ways to determine whether p is an outlier.

**Algorithm 1** Outliers Detection**Input:***δ*(*p*)                **Output:***Stat(tid)*, true for noise point1:**count NZPs:***n_NZP_* ← counter(*δ*(*p*))10:if *p* ∈ *H* then2:**if***n_NZP_* < *k*
**then**11:      **return:**
*Stat (tid)* ← *true*3:      **return:**
*Stat (tid)* ← *true*12:
**endif**
4:
**endif**
13:**calculate SD:** SD ← deviation(*H*)5:**ascending ordering:***H* ← sorter(*δ*(*p*))14:**if***SD* > *SD_max_*
**then**6:*nremain* ← *n_NZP_*15:      **return:**
*Stat (tid)* ← *true*7:
**do**
16:
**else**
8:      **remove outliers:** DixonTest(*H*, *n_remain_*)17:      **return:**
*Stat (tid)* ← *false*9:**while***k* < *nremain* < *n*18:
**endif**


①→④, for any p-centred k×k size window, when the number of NZP $n_{NZP}\left(\delta \left(p\right)\right)<k$, that is, there are only at most k−2 NZPs except p. As is shown in Figure 5, as an outlier, p locates on the jagged edges or in the blank area, and it can be directly identified as a noise point.①→②→④, the loop body in ② eliminates the outliers in the sample repeatedly until there are no more points that can be identified as an outlier. If p is eliminated, then it could be defined as an outlier.①→②→③→④, the remaining NZPs in ② will be sent to an SD calculator to make a macro evaluation of dispersion. p will be marked as an outlier if the SD is too large.

The improved algorithm not only checks whether the extreme point is an outlier, but the SD test is also carried out on the set after eliminating all outliers at the same time, which limits the dispersion of the cluster formed by remaining points. Thus, the macroscopic dispersion and the microscopic position distribution in the window could be simultaneously detected.

### 3.2. Edge Noise Filtering Approach

For depth images, the maximum depth gradient within a window should be limited by the line of sight. Thus, if any p is determined as an edge noise point, the maximum gradient in any p-centered window should be larger than a specific value.

In addition, this type of noise area can be eliminated by the approach which is shown in Figure 6. According to $n_{NZP}\left(\delta \left(p\right)\right)$, the determination of edge noise point will be based on the following three cases. 

$n_{NZP}\left(\delta \left(p\right)\right)<k$, when p locates at the jagged edges, p is defined as a noise point.$1.n_{NZP}\left(\delta \left(p\right)\right)=k$, if NZPs is arranged as a straight-line, p is defined as a noise point.$n_{NZP}\left(\delta \left(p\right)\right)>k$, in this case, fitting a plane π to G(δ(p)) by using the least-square method, where G(∗) is the converted 3D global coordinates of δ(p). Let S(G(δ(p)),π) be the SD of the vertical distances from G(δ(p)) to π, and α(π,λ) be the angle between π and sight plane λ. If $S\left(G\left(\delta \left(p\right)\right),\pi \right)>S_{\pi }$ or $\alpha \left(\pi ,\lambda \right)>\alpha_{p}$, define p as a noise point.

When $n_{NZP}\left(\delta \left(p\right)\right)\leq k$, the algorithm makes noise point judgments for the number and distribution of NZPs. In another case, the algorithm fits NZPs into a three-dimensional plane, and the decision is made by the angle of the plane and the dispersion of the distance between the plane and NZPs. $s_{p}$ and $\alpha_{p}$ are the given threshold values for judging p’s state. Selecting the appropriate threshold will help the system to separate the realistic area from the noise area. 

### 3.3. Plaque Noise Filtering Approach

Part of the plaque noise areas is a kind of residual noise, and they come from the residual part filtered by the above steps. Most of them are isolated and located in the areas that are supposed to be blank as an unrealistic surface. There are also some plaques connected to the realistic surface, which means that the serial algorithm for keeping the hand area by comparing the length of the chain code [32,33] will not always be effective. However, the use of a larger window is likely to cause excessive elimination of the hand area. Therefore, a method for orthogonally detecting the number of consecutive non-zero points is presented in Figure 7 to eliminate the plaque area. 

As shown in Figure 7, threads are opened for each $p\in I_{D}$ to search the adjoining continuous NZPs along rows and columns respectively, and obtain the numbers $n_{p_{x}}$ and $n_{p_{y}}$ which stand for how many NZPs are connected to p in the direction of rows and columns (including itself). By putting limits on $n_{p_{x}}$, $n_{p_{y}},$ and their product $m_{p},$ respectively, all plaque areas could be marked. The pseudo-code for one thread is shown in Algorithm 2.

**Algorithm 2** Plaque Noise Area Detection**Input:***ID*, *n_pymin_*, *n_pymin_*, *m_pmin_***Output:***Stat(tid)*, true for noise point1:**row search:** Xsearcher (*n_px_*, *I_D_*)2:**col search:** Xsearcher (*n_py_*, *I_D_*)3:*m_p_* = *n*_*px* ×_
*n*_*py*_4:**if***n_px_* < *n_pxmin_*||*n_py_* < *n_pymin_*||*m_p_* < *m_pmin_*
**then**5:      **return:**
*Stat (tid)* ← true6:
**else**
7:      **return:**
*Stat (tid)* ← false8:
**endif**


## 4. Experiments

To the best our knowledge, studies based on a non-smooth filter specially used for filtering the high noise point clouds generated by consuming depth cameras when used at close range have not been reported. Therefore, for comparison, we employed the standard median filter (SMF) and skin color based depth image classification (SCBDIC) (a fused method presented in [34,35]). Its principle is to register the depth and color image, and then remove the unrealistic surface from the point cloud by recognizing the skin color region. However, to obtain universal experimental results, all the experiments were carried out in an indoor fluorescent light environment, and the lighting conditions were not deliberately improved.

All experiments were conducted on a computer with an Intel Core i7 4770 @ 3.6 Ghz CPU and a Nvidia GTX 1060-6 GB graphics card. The depth sequence was captured using a Kinect v2.0 with resolution 512 × 424 at 30 fps and a SR300 with resolution 640 × 480 at 30 fps. The programming environment was Visual Studio 2017 with CUDA 9.2 version.

To prove the validity of our filtering system, we experimented with Kinect v2 and SR300 on hand depth images at different distances. Figure 8 shows the comparison of the proposed filtering system with the other two filters when using Kinect v2. A large amount of edge noise (marked by the blue circle) exists in the original depth image, which constitutes the unrealistic surfaces in the point cloud. However, the part of the color image corresponding to these unrealistic surfaces was not the skin color area, so they could be well removed by SCBDIC. However, the outliers and edge noise located inside the realistic surface (marked by the gray and black circle) could not be filtered out. More seriously, under different light and different angles, the colors had different changes, which can cause many hand areas (marked by the red circle) to be incorrectly recognized, resulting in over-filtering. On the contrary, SMF seems to be ineffective for such large noise areas, and can only filter out some outliers. The change in window size could not provide a better filtering effect, so we present the filtering effect of SMF when k=3 in Figure 8.

The proposed filter does not depend on other input sources. As long as the parameters are set reasonably, it can recognize almost all the noise areas and outliers then eliminate them. Even at a close distance of 500 mm, it still produces good results. By querying the judgment status of each sub-filtering algorithm, one type of noise area is not only recognized by its corresponding filtering algorithm, more often, it is recognized by both outlier filtering and edge noise filtering algorithms at the same time. This is because the region recognized as edge noise usually has high SD, which is also one of the characteristics of outlier noise. The reason why p is only recognized as edge noise is that the area where it is located is relatively smooth, but the angle between the fitted plane and the plane of view is too large. In addition, the reason why p is only recognized as an outlier is that in its window, the depth value of p is significantly different from other NZPs, and the values of other NZPs are not much different.

Since point clouds from SR300 hardly produce the realistic surfaces, the SMF that can filter out part of the outliers. Visually, this effect gets better as the filter window increases. As is shown in Figure 9, the edge noise area in orange circle and the outlier in gray circle are smoothed by SMF, and other noise areas in blue circles were also improved to some extent. However, at the same time, the gap between the two fingers (marked in red circle) was filled. At the same time, the depth values of almost all points were changed. This is equivalent to introducing a new error source. The proposed filtering system eliminated almost all edge noise regions without changing the depth value any point and preserved the raw depth data.

Figure 10 and Figure 11 show three views of the filtering effect of point clouds by using Kinect v2 and SR300 respectively. For two kinds of equipment with totally different noise characteristics, the proposed system can maintain a good filtering effect. It means that the proposed filtering system has certain universality by setting appropriate parameters. To get better results globally, the determination of the parameters of the filtering system requires a lot of experiments. We present the parameters used in the experiments in this paper which are listed in Table 1.

To evaluate the stability and real-time performance of the system, 1000 frames of continuous images were recorded as experimental material, and the run time for each frame was stable at 5 milliseconds, and the filtering effect video is updated as the Appendix A. Since all algorithms in this proposed filtering system adopt the parallel structure, the running speed of the system is not very sensitive to the resolution of depth image and more determined by GPU performance. 

At the same time, it also has excellent performance in stability. In Figure 12, 18 frames of different hand postures are shown, and most of them have very good filtering effects. However, it is noteworthy that, when a gradient of a part of the realistic surface gets too large, the system may determine it as an edge noise area and eliminate it. The frame in a red box indicates this situation.

## 5. Conclusions

When collecting depth images with a consumer depth camera, the noise interference becomes more serious as the object approaches. In order to eliminate these noise areas and obtain a correct pure raw point cloud with high resolution of the object, we proposed a new filtering system for using consumer depth cameras at close range in this paper. 

We classified the noise areas into three types, outlier noise, edge noise, and plaque noise. By analyzing the characteristics of these three noise types, we specially designed a filtering algorithm for each noise type: (1) an improved Dixon test algorithm for filtering outlier noise, (2) a three-dimensional plane fitting method to eliminate edge noise, and (3) an algorithm based on searching for the number of adjacent joints for the plaque noise. All algorithms adopted the parallel structure, which greatly improved the efficiency of the filtering system. The running speed of nearly 200 frames per second can meet the application of most real-time interactive systems. We tested the filtering system using two different depth cameras, and the filtering effects were much better than the other two filters involved in the comparison. This shows that the proposed filtering system has certain universality. At the same time, we also presented the system parameters that can achieve a better global filtering effect with the two cameras. Finally, in order to test the stability of the filtering effect, we used 1000 frame continuous hand depth images as experimental materials. The filtering effects show that the system can effectively eliminate most of the noise areas, and 18 of them were selected to present the filtering effect.

Excellent real-time, good filtering effect, and a certain degree of universality enables the proposed filtering system to be used as a pre-step for real-time human-computer interaction, real-time 3D reconstruction, and further filtering.

### Future Works

In the future, on the one hand, we will try to develop a method to evaluate the filtering effect, which can be used to realize the automatic optimization of system parameters, and increase or modify some sub-algorithms by using other kinds of cameras to improve the universality of the filtering system. On the other hand, we will try to develop a new algorithm that replaces the points that are marked as noise instead of simply removing them, making the filtered point cloud image edges smoother.

## Figures and Tables

**Figure 1 sensors-19-03460-f001:**
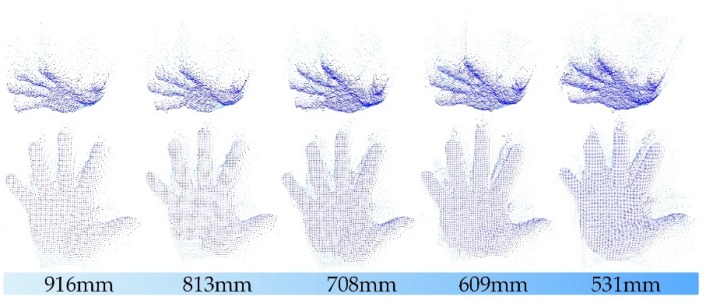
Point clouds of the human hand at different distances. The distance values are obtained by calculating the average of 25 × 25 pixels depth at the center of the hand.

**Figure 2 sensors-19-03460-f002:**
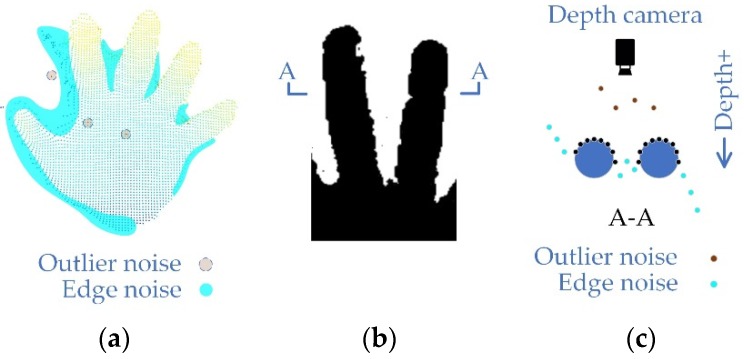
Noise classification. (**a**) A noisy point cloud image of the human hand at a distance of 800–900 mm. In order to show the details of the noise area, position A in (**b**) is cut and the section view is displayed in (**c**).

**Figure 3 sensors-19-03460-f003:**
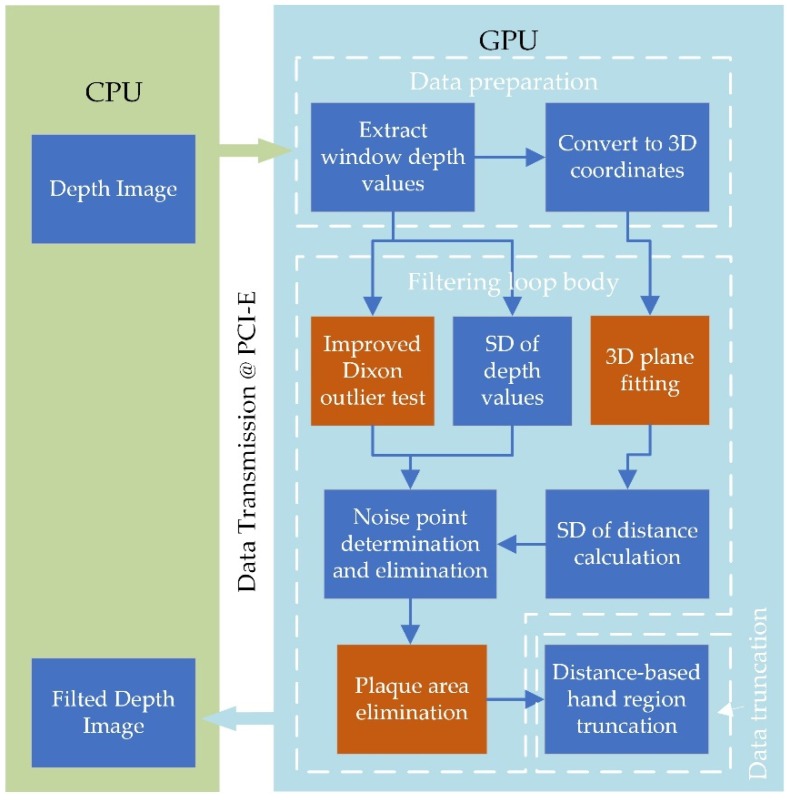
Proposed filtering system for hand depth image. The part with GPU algorithms of the system can be divided into three sub-parts. (1) Data preparation: extract the depth data of all points in the window corresponding to each thread, and work out their 3D coordinates. (2) Filtering loop body: as the core part of the filtering system, its main function is to identify the noise points and filter them out. (3) Data truncation: preserving the foreground and eliminating the depth image of the background. In addition, the three filtering algorithms proposed for different types of noise areas are marked in red boxes.

**Figure 4 sensors-19-03460-f004:**
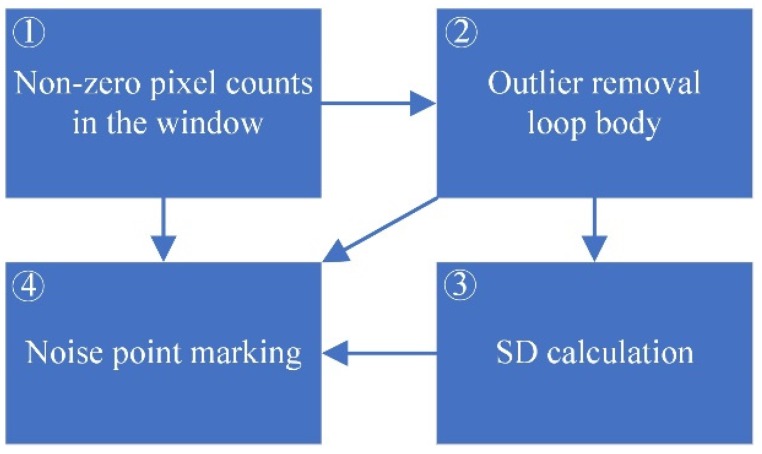
Improved Dixon test for depth image outlier detection. This algorithm combines the continuously draining outliers Dixon test and standard deviation (SD) calculation to perform macroscopic and microscopic identification of the center point.

**Figure 5 sensors-19-03460-f005:**
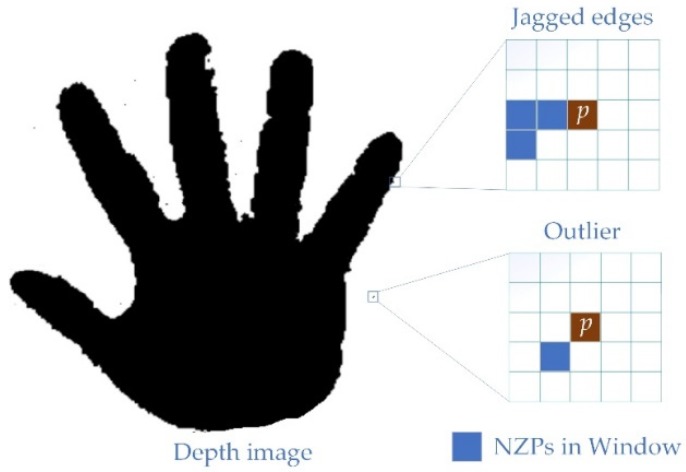
The locations of the points determined to be eliminated. For window areas with a small number of non-zero points, most of them are outliers or located on the jagged edge. This figure shows this case with *k* = 5. Other sizes of windows are similar.

**Figure 6 sensors-19-03460-f006:**
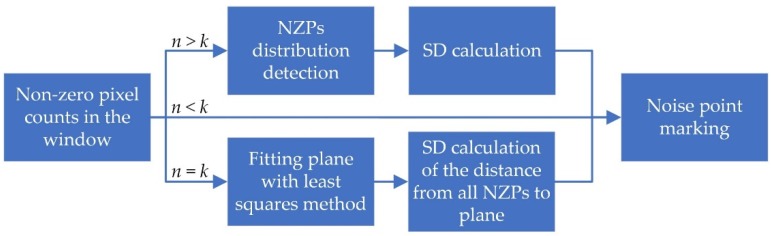
Edge noise filtering. Identification of this kind of noise points is mainly based on the relative position between non-zero points (NZPs).

**Figure 7 sensors-19-03460-f007:**
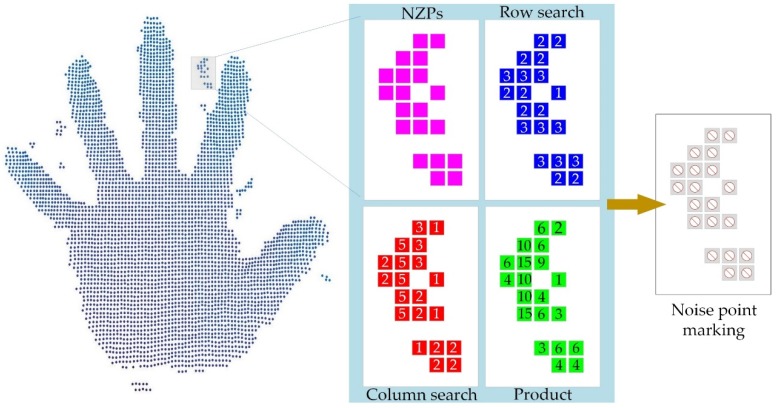
Orthogonally detecting the number of consecutive non-zero points of p.

**Figure 8 sensors-19-03460-f008:**
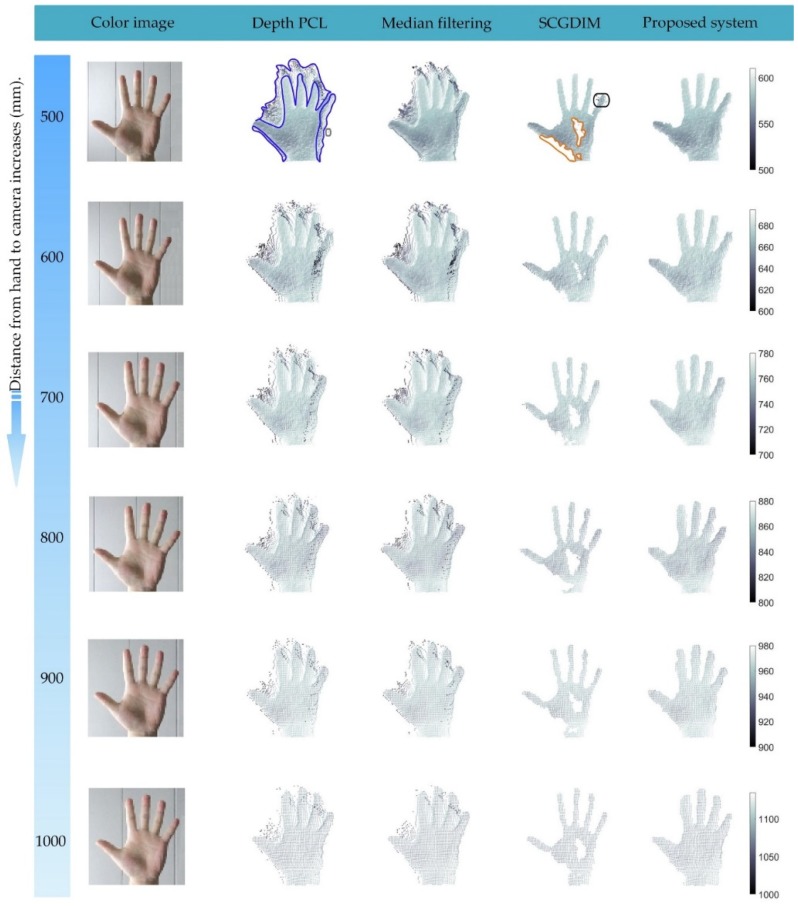
Comparisons at different distances by using Kinect v2.

**Figure 9 sensors-19-03460-f009:**
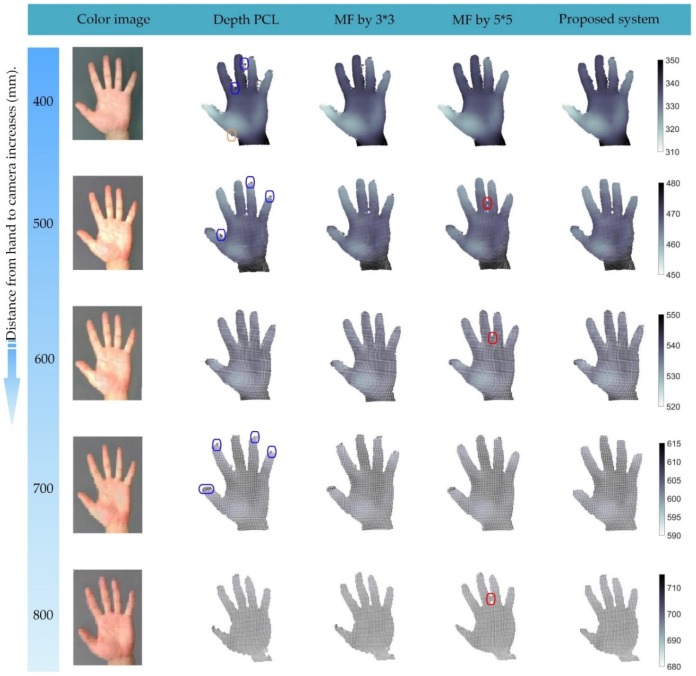
Comparisons at different distances by using SR300.

**Figure 10 sensors-19-03460-f010:**
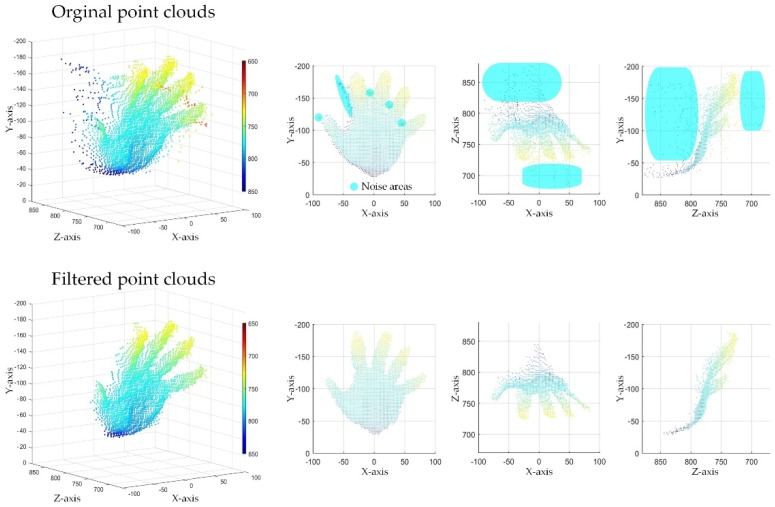
Three views of the filter result (Kinect v2).

**Figure 11 sensors-19-03460-f011:**
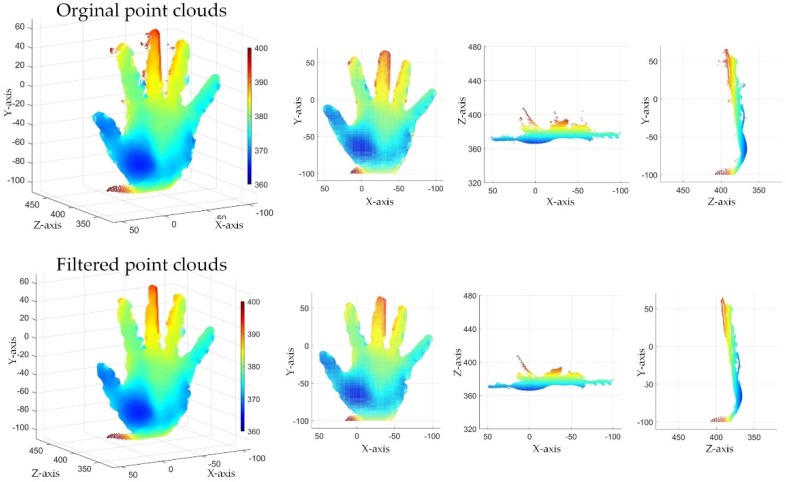
Three views of the filter result (SR300).

**Figure 12 sensors-19-03460-f012:**
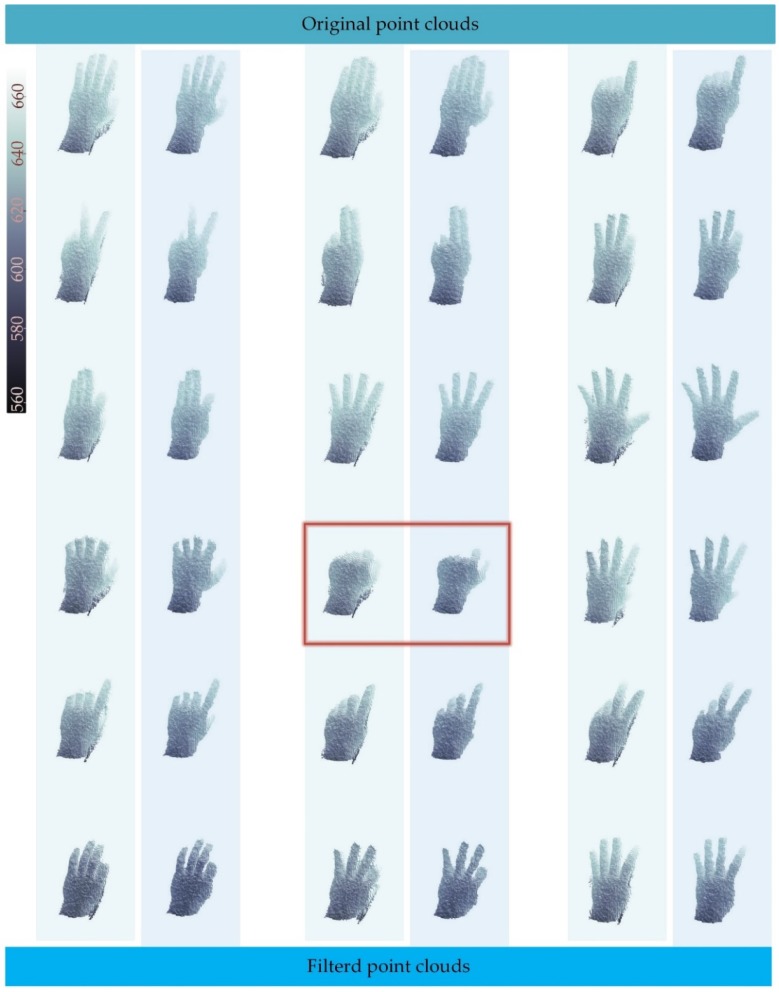
Comparisons of different hand gestures.

**Table 1 sensors-19-03460-t001:** Experimental parameters.

Dis.\Para.	k	nloop	α	Sp	npx	npy	mp	Sπ	αp
Kinect v2									
500 mm	3	20	0.1000	4.3	7	7	25	3	75
600 mm	3	20	0.1000	4.55	5	5	25	3	75
700 mm	3	15	0.1000	4.8	5	5	25	3	75
800 mm	3	15	0.1000	5.05	4	4	16	3	75
900 mm	3	10	0.1000	5.3	4	4	16	3	75
1000 mm	3	10	0.1000	5.55	4	4	16	3	75
SR300									
400 mm	3	10	0.1000	2.5	5	5	25	1.5	70
500 mm	3	10	0.1000	2.5	5	5	25	1.5	70
600 mm	3	10	0.1000	2.3	5	5	25	1.2	70
700 mm	3	10	0.1000	2.2	4	4	16	1	70
800 mm	3	10	0.1000	2.2	4	4	16	1	70

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
