# Peer review of "A New Filtering System for Using a Consumer Depth Camera at Close Range"

_sensors, 2019, doi:10.3390/s19163460_

Round 1

Reviewer 1 Report

My main concern is that your paper is devoted to a device that is not in the market anymore so your paper will be of no interest to most people. Please, do experiments with SR300 and (or else) the ZED camera data and you will have a very interesting paper that will be better for the udience of Sensors. Also, make your conclusion better (as indicated below).

1) "According to the author's observation": why don´t you use first person plural; sounds better in situations like this one. So change here (and all places) to "According to our observation" if you wish.

2) "the Intel SR300 also has a similar situation." I was going to suggest you to generalize and not depict it in details for the case of Kinect. So if it is applicable to SR300, I would like to see an application of it in the next paper version. The main reason is because Kinect died, as you know it. Pick up some data from Internet for this.

3) "could eliminate of vast majority" -> "could eliminate the vast majority"

4) "In addition, this article only studies the noise of Kinect 71 v2 generated in Windows SDK" (that is not made anymore; that is my major concern; nobody is buying the kinect so your method lost or will lost interest soon). I known (if you want to publish) that you should make more experiments in order to show the method versatility. You could take some data sample from the web, e.g.: http://www.nlpr.ia.ac.cn/iva/yfzhang/datasets/egogesture.html (might you have better ones for SR300) please do it. In fact, you could generalize your method with these more experiments, got it.

5) I did not understand this piece (missing some thing): "For a highly integrated device, the calibration method for laser scanner and cannot be used [27]"

6) "this results in the user cannot adjust" -> "and as a result the user cannot adjust"

7) The paper should go to an English revision. Several constructions are not OK, as "where has a large gradient", "and meets the", 

8) The paper should be selfcontained, that is, several terms are not defined previous to use as "outliers", "realistic surface" (that is different from real or actual surface), "salt-and-pepper noise", 

9) Your conclusions are so weak. Add to it more text, saying at general (one paragraph for each of these): what did you do, how did you do it, what are the main resuts, what are the contributions, what are the applications and what do you intend to do as future research (with ideas on how to do it). Most fo these are there (in a very short way) however you can separate them, then you would have a much better conclusion.

Reviewer 2 Report

The paper introduces a methodology of a real-time filtering system for eliminating noise areas in close range applications when utilizing the Kinect v2 sensor. The topic is interesting for the readers of Sensors. However, the manuscript has important weaknesses and it is not well structured.

It is not clearly presented which are the close range applications that could be benefited from the introduced methodology. This becomes evident right from the beginning as the introduction starts with the utilized sensor. It needs to be reorganized and clearly state the problem. Other research approaches should also be clearly articulated and the proposed methodology should follow later in the text.

A major shortcoming of the paper is that the proposed methodology is not validated. I would recommend scanning objects with known dimensions, under different lighting conditions, in order to validate in a quantitative manner the introduced methodology.

Another weakness is that the experimental results are discussed briefly. A thorough discussion of the main findings should be presented.

Furthermore, the authors must proofread the manuscript as it includes some minor errors and misspells.

Some more specific comments are:

Abstract

Line 11: Please avoid such statements „… is a disaster …“

Line 17: It is more important to mention the utilized methodologies and not what was not used.

Line 18: The statement “good results” is rather vague. Please be more specific (provide numbers).

Introduction

Figure 1 is not suitable for the Introduction as this is part of the obtained results.

Kinect Noise Characterization

Lines 77-79: This sentence is very confusing. Please rewrite it.

Figure 2 should be better described in the text.

Figure 3 should be better described in the text. Also, indicate the 3 sub-parts in figure 3.

Experiments

Line 214: Mention the other two filters.
